# Tuberculosis treatment outcomes and associated factors at Alemgena Health Center, Sebeta, Oromia, Ethiopia

**Kedija Hayre[1]◉, Mihiretu Kumie Takele[1]◉, Dagim Jirata Birri◉[2]◉ \***

**1** Department of Public Health, Ayer Tena Health Science and Business College, Addis Ababa, Ethiopia,
**2** Department of Microbial, Cellular and Molecular Biology, Adids Ababa University, Addis Ababa, Ethiopia

◉ These authors contributed equally to this work.
\* dagimj@yahoo.com

## Abstract

### Background

Tuberculosis (TB) is a global public health problem. Evaluation of TB treatment outcome enables health institutions to measure and improve the effectiveness of TB control programs. This study aimed to assess treatment outcomes of tuberculosis and identify associated factors among TB patients registered at Alemgena Health Center, Oromia, Ethiopia.

### Method

A retrospective study was conducted; Secondary data were collected from medical records of 1010 TB patients treated at Alemgena Health Center between September 2012 and August 2018, inclusively. Logistic regression was used to identify factors associated with TB treatment outcomes. P-value less than 0.05 was considered statistically significant.

### Results

The proportion of males and females was almost equal. Among the patients 64.7% were in the age group 15–34, 98% were new cases, 31.2% were smear positive, 13% were HIV positive and 40.3% had extra-pulmonary tuberculosis. 94.2% of the patients had successful treatment outcome, with 26.9% cured and 67.3% treatment completed, whereas 5.8% had unsuccessful treatment outcomes, of whom 4.2% died and 1.5% defaulted. Death rate was higher among patients older than 44 years (10.4%) than among children (0%). In bivariate logistic regression analysis, treatment success rate was 3.582 (95% CI 1.958–6.554, p-value = .000) times higher in the age group 44 and below compared to the age group 45 and above.

### Conclusion

Treatment success rate exceeded the one targeted by WHO. Age was found to be associated with treatment outcome. Success rate has to be improved for TB patients in the age group greater than 45 years of age.

**Data Availability Statement:** "All relevant data are within the paper and its supporting information files."

**Funding:** The author(s) received no specific funding for this work.

**Competing interests:** The authors have declared
that no competing interests exist.

## Introduction

Tuberculosis (TB) is a bacterial infectious disease caused principally by *Mycobacterium tuberculosis* (MTB), and less commonly by *Mycobacterium bovis*. Humans are the only natural reservoir for MTB [1]. Although the principal body site affected is the lung (pulmonary tuberculosis), other body parts can be involved (extra-pulmonary tuberculosis). MTB is transmitted from infected persons to health persons predominantly by inhalation of droplet nuclei aerosolized by coughing, sneezing, talking, laughing, or singing. Crowded living conditions facilitate the spread of TB. Zoonotic tuberculosis, caused by *M. bovis*, can be transmitted from cattle to humans through animal contact and consumption of unpasteurized milk [2].

One-quarter of the world's population is infected with MTB; however, a small fraction of infected persons develop active tuberculosis disease in their life time [3]. The risk factors for the development of active TB from MTB infection is infection with acquired immunodeficiency syndrome (AIDS), under-nutrition, diabetes, smoking and alcohol consumption [3–5]. About 90% of those who develop TB are adults, men being more affected than women [3]. Globally, men accounted for 55%, women for 33% and children for 12% of TB burden in 2022 [6].

Tuberculosis is one of the major public health problems worldwide. It is the 13th leading cause of death globally [3]. It is the leading cause of death resulting from a single infectious agent. It is also the leading cause of death in people living with HIV and death related to antimicrobial resistance [7]. According to the World Health Organization (WHO) report, in 2022, an estimated 10.6 million new tuberculosis cases and 1.3 million deaths due to tuberculosis occurred worldwide and COVID 19 pandemic increased the number of TB related deaths [6]. In 2022, 6.3% of the new TB cases were people living with HIV. Among the global TB deaths that occurred in HIV negative persons in 2022, 52% were men, 32% were women and 16% were children [6]. The sub-Saharan African accounted for 23% of global TB cases [6] and for 68.3% of TB deaths in HIV-positive cases in 2022 [6]. Among those who fell ill with tuberculosis in 2022, 55% were adult men, 33% were adult women and 12% were children. About 1.13 million and 167, 000 deaths occurred among HIV negative and HIV positive persons, respectively [6]. In 2022, men, women and children accounted for 47%, 35% and 18% of global TB deaths among HIV-positive persons, respectively [6]. World Health Organization estimates that about 5,000 tuberculosis related deaths occur daily globally, of which, 95% take place in low- and middle-income countries. Africa is the second TB burden region (25%) next to south-east Asia (44%) [6]. In 2022, Ethiopia ranked 19th in the 30 countries that had the highest burden of TB [6]. There were estimated 156 000 new tuberculosis cases and 22700 deaths (1700 deaths occurred among HIV positive cases) in Ethiopia in 2022 [6]. Thus Ethiopia accounted for about 17.5% of global tuberculosis mortality in 2022.

Tuberculosis is a treatable and curable disease. In the absence of treatment with effective antiTB drugs about 70% of smear positive patients die within 10 years [8]. The currently recommended treatment for people with drug-susceptible TB disease is a 6-month regimen of four first-line drugs: isoniazid, rifampicin, ethambutol and pyrazinamide. The advent of modern anti-mycobacterial drug therapy has made the treatment of tuberculosis successful. Effective treatment is one of the crucial measures in the control of tuberculosis. In Ethiopia, drug-susceptible tuberculosis patients are treated with four drugs (isoniazid, rifampin, pyrazinamide, and ethambutol) in the first two months, followed by administration of two drugs (isoniazid and rifampin) in the next four months. Tuberculosis treatment effectiveness may be hampered due to drug resistance or an inappropriate regimen, or because of nonadherence of patients to antiTB drug therapy.

About 8% of TB patients can be successfully treated with a 6-month drug regimen [6]. In 2021, the treatment success rate for people treated for TB with first-line regimens was 88%

globally. Treatment success rates remain lower among people living with HIV (79% globally in 2021). The treatment success rate for children was 91% in 2021. Treatment outcomes measure the effectiveness of national TB control programs [9]. This can be true for health institutions such as health centers and hospitals engaged in TB treatment. According to WHO, an effective TB control program should achieve at least 90% treatment success rate and 85% cure rate [10]. Achieving these targets significantly contributes to the reduction of tuberculosis spread at household and community levels, and to the reduction of complications and mortality that could result from tuberculosis. Many factors, such as HIV, age, under-nutrition, diabetes, TB disease severity, extra-pulmonary TB, history of TB, adherence, alcohol use and adverse drug reactions, have been shown to be associated with unsuccessful treatment outcomes TB, although results vary by setting and patient population [11–14].

TB treatment outcome in Ethiopia can vary from one setting to another. There are not published research reports on TB treatment outcome in Alemgenea area. Evaluation of TB treatment outcome enables health institutions to measure and improve the effectiveness of TB control programs. The aim of the this study was, therefore, to assess the treatment outcome of drug-sensitive TB and to identify associated factors among TB patients who got health care service at Alemgena Health Center over a period of 6 years from September 2012 to August 2018.

## Materials and methods

### Study area

The study was conducted at Alemgena Health Center, which is located in Sebeta district in Sheger city of Oromia regional state, Ethiopia. It provides health care service for patients in Sebeta district and Alemgena town.

### Study design, population and sampling

A 5 years retrospective study was employed to achieve the objective of this study. Data collection was conducted in May 2019.

The source population was medical records of all TB patients who were on anti-tuberculosis treatment at TB clinic of Alemgena Health Center. The population of the study was medical records of all TB patients who were on anti-tuberculosis treatment at TB clinic of Alemgena Health Center from September 2012 to August 2018.

Medical records of all TB patients who were on anti-tuberculosis treatment from 2012 to 2018 were included in the study. There were a total of 1010 patients in the TB register.

### Data collection

A format prepared for secondary data collection was used to gather TB patients' information on the sociodemographic variables and TB related variables. These included patients' age, sex, body weight, residence, TB type, TB category, treatment outcomes and HIV status.

### Dependent and independent variables

Dependent variables included tuberculosis treatment outcomes, whereas the independent variables consisted of age, sex, weight, TB/HIV co-infections, and type of TB.

### Inclusion criteria and exclusion criteria

All TB patients' record in the TB clinic from September 2012 to August 2018 were included in the study. TB patients with incomplete records were excluded.

## Data processing and analysis

Data were checked for completeness, cleaned, coded, entered and analyzed using SPSS version 20. A descriptive analysis was conducted to get summary values of variables. Logistic regression analysis was used to identify factors associated with treatment outcomes. P-value less than 0.05 was considered statistically significant.

## Ethical considerations

Ethical approval was obtained from ethical review board of Ayer Tena Health Science and Business College. Moreover, Alemgena Health Center permitted us to use the records of TB patients for the purpose of this study, following our written request for permission. Secondary data (medical records of TB patients), which were accessed on April 19, 2019, was used for this study and we did not have contact with human participants. So we did not have to obtain informed consent. During secondary data collection, we did not record names of patients for the sake of confidentiality.

## Definitions of terms

**Cured.** A pulmonary TB patient with bacteriologically confirmed TB at the beginning of treatment who was smear or culture negative in the last month of treatment and on at least one previous occasion.

**Treatment completed.** A TB patient who completed treatment without evidence of failure but with no record to show that sputum smear or culture results in the last month of treatment and on at least one previous occasion were negative, either because tests were not done or becuase results are unavailable.

**Treatment failed.** A TB patient whose sputum smear or culture is positive at month 5 or later during treatment.

**Defaulted.** A TB patient whose treatment is interrupted for two consecutive months or more.

**Died.** A TB patient who died for any reason before starting or during the course of treatment.

**Treatment success.** The sum of cured and treatment completed.

**Treatment unsuccess.** The sum of treatment failed, defaulted and died.

**Transferred out.** A patient who moved to a different health institution.

## Results

### Socio-demographic characteristics of study participants

A total 1010 records of TB patients were included in this study. The proportion of males and females is almost equal (Table 1). 64.7% (641/990) of the patients were in the age group 15–34. TB patients aged 55 years and above were the least (4.45%). The mean age was 30.50 (standard deviation = 14.71). 89.9% (889/990) of the patients were in the age group $\leq$ 50, with the age group 15–45 comprising 86% (853/990). The minimum and maximum age was 1 year and 89 years, respectively, giving a range of 88 years. The majority of the TB patients in the age group $\leq$ 29 were females (53.5%; 318/594), and males in the age group 50 and above (63.8%; 74/116). However, for the age group 30–49, the distribution of males and females was more or less equal (50.9% females and 49.1% males) (P value = 0.03). Data on sex and age were not available for 20 of the patients. The above analyses excluded these missing data on sex and age.

**Table 1. Socio-demographic and clinical characteristics of TB patients (n = 1010).**

| Variable | Category | Frequency | Percent |
|---|---|---|---|
| **Sex (n = 990)** | Male | 488 | 49.3 |
| | Female | 502 | 50.7 |
| **Age (n = 990)** | 0–14 | 67 | 6.8 |
| | 15–24 | 312 | 31.5 |
| | 25–34 | 329 | 33.2 |
| | 35–44 | 119 | 12.0 |
| | 45–54 | 74 | 7.5 |
| | 55–64 | 44 | 4.4 |
| | 65+ | 45 | 4.5 |
| **Smear result (n = 629)** | Positive | 196 | 31.2 |
| | Negative | 433 | 68.8 |
| **Patient category (n = 928)** | New | 909 | 98 |
| | Relapse | 15 | 1.6 |
| | Transfer in | 4 | 0.4 |
| **Site of TB (n = 982)** | Pulmonary positive | 263 | 26.8 |
| | Pulmonary negative | 324 | 33 |
| | Extra-pulmonary | 395 | 40.2 |
| **HIV test result (n = 943)** | Reactive | 123 | 13.0 |
| | Nonreactive | 820 | 87.0 |
| **Treatment outcome (n = 862)** | Cured | 232 | 26.9 |
| | Completed | 580 | 67.3 |
| | Subtotal successful | 812 | 94.2 |
| | Failed | 1 | 0.1 |
| | Died | 36 | 4.2 |
| | Defaulted | 13 | 1.5 |
| | Subtotal unsuccessful | 50 | 5.8 |
| **Year of registration (n = 1010)** | 2012 | 68 | 6.7 |
| | 2013 | 226 | 22.4 |
| | 2014 | 201 | 19.9 |
| | 2015 | 146 | 14.5 |
| | 2016 | 167 | 16.5 |
| | 2017 | 152 | 15.0 |
| | 2018 | 50 | 5.0 |

## Clinical characteristics

As Table 1 shows, among the TB patients, 98% were new cases, 31.2% (196/629) were smear positive, 13% were HIV positive and 40.3% (397/984) were diagnosed with extra-pulmonary tuberculosis. The highest (16.6%) TB-HIV co-infection occurred in 2014 and the lowest (9.5%) in 2013. The proportion of extra-pulmonary TB cases increased form 36.8% in 2014 to 46.3% in 2017. Data on patient category, smear result, TB type and HIV status were not available for 82, 381, 27 and 67 of the patients, respectively. The above analyses excluded these missing data.

## Treatment outcomes

Regarding TB treatment outcome (Table 1), 94.2% of the patients had successful treatment outcome, with 26.9% cured and 67.3% treatment completed. 5.8% had unsuccessful

**Table 2. Treatment outcome by demographic and clinical characteristics.**

| Variables | Response | Treatment outcome, n(%) | | | | | | | Total |
|---|---|---|---|---|---|---|---|---|---|
| | | Successful | | | Unsuccessful | | | | |
| | | Cured | Completed | Subtotal | Failed | Died | Default | Subtotal | |
| **Sex** | Male | 120(29.1) | 268(65.0) | 388(94.2) | 0(0) | 16(3.9) | 8(1.9) | 24(5.8) | 412(48.8) |
| | Female | 103(23.8) | 303(70.1) | 406(94.0) | 1(0.2) | 20(4.6) | 5(1.2) | 26(6) | 432(51.2) |
| | Total | 223(26.4) | 571(67.7) | 794(94.1) | (0.1) | 36(4.3) | 13(1.5) | 50(5.9) | 844(100) |
| **Age** | ≤ 14 | 3(5.2) | 54(93.1) | 57(98.3) | 0(0) | 0(0) | 1(1.7) | 1(1.7) | 58(6.9) |
| | 15–24 | 80(29.3) | 178(65.2) | 258(94.5) | 0(0) | 10(3.7) | 5(1.8) | 15(5.5) | 273(32.3) |
| | 25–34 | 82(29.8) | 181(65.8) | 263(95.6) | 1(0.4) | 9(3.3) | 2(0.7) | 12(4.4) | 275(32.6) |
| | 35–44 | 31(30.1) | 69(67) | 100(97.1) | 0(0) | 3(2.9) | 0(0) | 3(2.9) | 103(12.2) |
| | 45–54 | 11(18.6) | 41(69.5) | 52(88.1) | 0(0) | 6(10.2) | 1(1.7) | 7(11.9) | 59(7.0) |
| | 55–64 | 7(18.9) | 24(64.9) | 31(83.8) | 0(0) | 4(10.8) | 2(5.4) | 6(16.2) | 37(4.4) |
| | 65+ | 9(23.1) | 24(61.5) | 33(84.6) | 0(0) | 4(10.3) | 2(5.1) | 6(15.4) | 39(4.6) |
| | Total | 223(26.4) | 571(67.7) | 794(94.1) | 1(0.1) | 36(4.3) | 13(1.5) | 50(5.9) | 844(100) |
| **Smear result** | Positive | 143(81.2) | 26(14.8) | 169(96) | 0(0) | 5(2.8) | 2(1.1) | 7(4) | 176(31.8) |
| | Negative | 22(5.8) | 339(89.9) | 361(95.8) | 0(0) | 15(4) | 1(0.3) | 16(4.2) | 377(68.2) |
| | Total | 165(29.8) | 365(66) | 530(95.8) | 0(0) | 20(3.6) | 3(0.5) | 23(4.2) | 553(100) |
| **TB Type** | PTB positive | 184(79.7) | 34(14.7) | 218(94.4) | 0(0) | 10(4.3) | 3(1.3) | 13(5.6) | 231(27.5) |
| | PTB negative | 21(8.0) | 222(84.4) | 243(92.4) | 1(0.4) | 15(5.7) | 4(1.5) | 20(7.6) | 263(31.3) |
| | EPTB | 16(4.6) | 312(90.4) | 328(95.1) | 0(0) | 11(3.2) | 6(1.7) | 17(4.9) | 345(41.1) |
| | Total | 221(26.3) | 568(67.7) | 789(94.0) | 1(0.1) | 36(4.3) | 13(1.5) | 49(5.8) | 839(100) |
| **HIV status** | Positive | 26(26.0) | 70(70.0) | 96(96.0) | 0(0) | 4(4.0) | 0(0) | 4(4.0) | 100(12.3) |
| | Negative | 191(26.8) | 482(67.6) | 672(94.4) | 1(0.1) | 27(3.8) | 12(1.7) | 40(5.6) | 712(87.9) |
| | Total | 217(26.7) | 551(67.9) | 768(94.6) | 1(0.1) | 31(3.8) | 12(1.2) | 44(5.4) | 812(100) |
| **Year** | 2012 | 14 (25) | 36(64.3) | 50(89.3) | 0(0) | 3(5.4) | 3(5.4) | 6(10.7) | 56(6.5) |
| | 2013 | 49(27.1) | 113(62.4) | 162(89.5) | 0(0) | 9(5.0) | 10(5.5) | 19(10.5) | 181(21.0) |
| | 2014 | 42(26.8) | 105(66.9) | 147(93.6) | 1(0.6) | 9(5.7) | 0(0) | 10(6.4) | 157(18.2) |
| | 2015 | 31(25.8) | 87(72.5) | 118(98.3) | 0(0) | 2(1.7) | 0(0) | 2(1.7) | 120(13.9) |
| | 2016 | 40(25.6) | 112(71.8) | 152(97.4) | 0(0) | 4(2.6) | 0(0) | 4(2.6) | 156(18.1) |
| | 2017 | 41(28.8) | 94(65.7) | 135(94.4) | 0(0) | 8(5.6) | 0(0) | 8(5.6) | 143(16.6) |
| | 2018 | 15(30.6) | 33(67.3) | 48(98.0) | 0(0) | 1(2.0) | 0(0) | 1(2.0) | 49(5.7) |
| | Total | 232(26.9) | 580(67.3) | 812(94.2) | 1(0.1) | 36(4.2) | 13(1.4) | 50(5.8) | 862(100) |

PTB = pulmonary tuberculosis; EPTB = extra-pulmonary tuberculosis

treatment outcomes, of whom 4.2% died and 1.5% defaulted. Data on treatment outcome of 66 patients was not available and 82 patients transferred out. Therefore, in treatment outcome analysis, we excluded a total of 148 (66+82) patients. Details of treatment outcome by demographic and clinical characteristics is shown in Table 2. Treatment success rate was 94% for female and 94.2% for male TB patients. Treatment success rate for children (aged ≤ 14) was 98.3%, but it was 84.2% for patients older than 54 years of age. Death rate was higher among patients older than 44 years of age (10.4%) than among children (0%). A similar treatment success rate was found among smear positive (96%) and smear negative (95.8%) TB patients. In addition, treatment success rate was somewhat similar among pulmonary positive (94.4%), pulmonary negative (92%) and extra-pulmonary (95.1%) TB patients. HIV positive patients and HIV negative TB patients had a treatment success rate of 96% and 94.4%, respectively.

**Table 3. Bivariate logistic regression analysis of factors associated with TB treatment outcome.**

| Variables | Category | Treatment outcome, n(%) | | | COR [95% CI] | P-value |
|---|---|---|---|---|---|---|
| | | Successful | Unsuccessful | Total | | |
| **Sex** | Male | 388(94.2) | 24(5.8) | 412 | 1.035 (0.584–1.834) | 0.905 |
| | Female | 406(94) | 26(6) | 432 | | |
| **Age** | ≤ 44 | 678(95.6) | 31(4.4) | 709 | **3.582 (1.958–6.554)** | **0.000** |
| | 45+ | 116(85.9) | 19(14.1) | 135 | | |
| **Smear result** | Positive | 169(96) | 7(4) | 176 | 1.070 (0.432–2.650) | 0.884 |
| | Negative | 361(95.8) | 16(4.2) | 377 | | |
| **TB type** | Plumnory TB | 461(93.3) | 33(6.7) | 494 | 0.722 (0.395–1.318) | 0.289 |
| | Extra-plumonary TB | 328(95.1) | 17(4.9) | 345 | | |
| **HIV test result** | Positive | 96(96) | 4(4) | 100 | 1.429 (0.500–4.082) | 0.505 |
| | Negative | 672(94.4) | 40(5.6) | 712 | | |

### Factors related to treatment outcome

Our study showed that there is statistically significant association between TB treatment outcomes and age (P = 0.001) (Table 3). In bivariate logistic regression analysis, treatment success rate was 3.582 (95% CI, 1.958–6.554; p-value = .000) times higher in the age group ≤ 44 compared to the age group 45 and above. Bivariate analysis did not reveal significant association between TB treatment outcome and other factors such as sex, TB type, smear result and HIV test result.

## Discussion

This study attempted to assess tuberculosis treatment outcome at Alemgenea Health Center. The study revealed that the overall treatment success rate was very high (94.4%) and exceeded the target of WHO. This is similar to the figure reported from Dilla and Bale Goba Referral hospitals in Ethiopia, in which the success rate was 92.3% [15] and 91.2% [16], respectively. However, it is higher than that of Adam city private and public health facilities (80.8%) [17] and Wolayta Sodo referral Hospital in Ethiopia (82.5%) [18], Atwima Nwabiagya District in Ghana (68.46%) [19], Lagos private hospitals (78.09%), a tertiary hospital (75.3%) [20], another tertiary hospital in a former in Ebony state (56.5%) [21], Jos-North Mongu (Plateau State) and Anambra and Oyo states (75.8%) in Nigeria (67.4%) [22–24], rural eastern Uganda (81.1%) [25], Malaysia (80.7%) [26] and Finland (75%) [27]. This discrepancy may be explained by low follow-up in developing countries and by a relatively higher death rate (16%) among old TB patients in the case of TB patients from Finland [27].

The finding that the death rate of TB patients in the current study is 4.2% is comparable to that of TB patients of various health institutions in Ethiopia, including Wolayta Sodo referral Hospital (4.7%) [18], Adam city private and public health facilities (5%) [17] and Madda Walabu University Goba Referral Hospital (5.1%) [16]. But it is lower than the figure reported from a tertiary hospital in a former Ebony state of southeast Nigeria (14%) [21] and Malaysia (10.2%) [26]. This difference may be attributed to high level of HIV co-infection (29.5%) and diabetes mellitus (18.3%) for the Nigerian and Malayian study, respectively, which could result in increased death rate.

Several studies have shown that many factors such as HIV status, age and site of tuberculosis are associated with TB treatment outcome [15, 16, 23, 25, 26, 28]. In the present study, the associated factor is age alone and this is similar to the findings from Madda Walabu University Goba Referral Hospital [16], Malaysia [26], Nigeria [23], Kilifi country of Kenya [29], Uganda

[25] and Hunan state of China [30]. The finding that TB treatment success rate was lower in patients older than 44 years is in agreement with study conducted on TB patients of Gambella regional state of Ethiopia [31]. The lower success rate in the relatively older age group could be explained by gradual decrease of cellular immunity that plays a vital role in combating the intracellular pathogen *Mycobacterium tuberculosis.*

This study has several limitations. It is based on analysis of secondary data, which did not include many socio-demographic and clinical factors such as income, education level, marital status, occupation, comorbidities and others that might have effect on TB treatment outcome. Moreover, the secondary data may have poor quality. Therefore, the findings have to be interpreted carefully after taking these limitations into account.

## Conclusions

The TB treatment success rate in the current study surpassed the 90% treatment success rate set by WHO. Treatment outcome was significantly associated with age. Success rate has to be improved for TB patients older than 45 years of age through, for example, strict follow-up of patients.

## Supporting information

**S1 Data.**
(SAV)

## Acknowledgments

We thank the medical director (Dawit Besha) and TB clinic staff of Alemgena Health Center for permitting and helping us to use data by providing patient records.

## Author Contributions

**Conceptualization:** Kedija Hayre, Mihiretu Kumie Takele, Dagim Jirata Birri.

**Data curation:** Kedija Hayre, Dagim Jirata Birri.

**Formal analysis:** Dagim Jirata Birri.

**Investigation:** Kedija Hayre, Mihiretu Kumie Takele, Dagim Jirata Birri.

**Methodology:** Kedija Hayre, Mihiretu Kumie Takele, Dagim Jirata Birri.

**Supervision:** Mihiretu Kumie Takele, Dagim Jirata Birri.

**Writing – original draft:** Dagim Jirata Birri.

**Writing – review & editing:** Mihiretu Kumie Takele, Dagim Jirata Birri.

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
