## [Decision Letter · Decision Letter 0]

28 Dec 2023

PONE-D-23-37113Tuberculosis Treatment Outcomes and associated factors at Alemgena Health Center, Sebeta, Oromia, EthiopiaPLOS ONE

Dear Dr. Birri,

Thank you for submitting your manuscript to PLOS ONE. After careful consideration, we feel that it has merit but does not fully meet PLOS ONE’s publication criteria as it currently stands. Therefore, we invite you to submit a revised version of the manuscript that addresses the points raised during the review process.

Please submit your revised manuscript by Feb 11 2024 11:59PM. If you will need significantly more time to complete your revisions, please reply to this message or contact the journal office at plosone@plos.org. Please include the following items when submitting your revised manuscript:A rebuttal letter that responds to each point raised by the academic editor and reviewer(s). You should upload this letter as a separate file labeled 'Response to Reviewers'.A marked-up copy of your manuscript that highlights changes made to the original version. You should upload this as a separate file labeled 'Revised Manuscript with Track Changes'.An unmarked version of your revised paper without tracked changes. You should upload this as a separate file labeled 'Manuscript'.

We look forward to receiving your revised manuscript.

Kind regards,

Frederick Quinn

Academic Editor

PLOS ONE

Journal Requirements:

2. In the online submission form, you indicated that Data will be obtained from the corresponding author up on request.

Reviewers' comments:

Reviewer's Responses to Questions

**Comments to the Author**

1. Is the manuscript technically sound, and do the data support the conclusions?

Reviewer #1: Partly

Reviewer #2: Yes

2. Has the statistical analysis been performed appropriately and rigorously? 

Reviewer #1: No

Reviewer #2: Yes

3. Have the authors made all data underlying the findings in their manuscript fully available?

Reviewer #1: Yes

Reviewer #2: Yes

4. Is the manuscript presented in an intelligible fashion and written in standard English?

Reviewer #1: No

Reviewer #2: No

5. Review Comments to the Author

Reviewer #1: General comments: This manuscript has generated important data for TB control program in local context. However, it has limited global implication. If the authors re-write the manuscript including re-analysis it can be considered for publishing in Plos One.

Detail comments as follow;

Abstract

Result: Line 16-19: Please, avoid ambiguous words like about, nearly, almost all. Such words are frequently used even in the body of manuscript

Introduction:

Line 31-32: ---------. Humans are the only

reservoir for MTB. There are number recent report on MTB infections in animals how authors dare to say human are the only reservoir for MTB? Can authors justify?

Line 30-38: whole paragraph has no reference citation

Authors used almost same source of information, WHO report and advised to use scientific report/journal articles rather than using report. Moreover, authors used different citation style in one paper. For instance, author, date (WHO, 2021) and number in bracket (2). Authors should use Plos One reference style

Materials and Methods

Study design: Line 109: typo error . A 5 years………..was employed to to achieve……

Line: 111-121: Authors advised to organize under one sub-heading.

Ethical consideration

Although authors obtained ethical approval for study form ethical review board of Ayer Tena Health Science and Business College, they didn’t mention about consent of Alemgena health centre to give them permission to use their data for this study.

Results: Line 168-169: (standard deviation = 14.71). Nearly 90% (889/990) of the patients were in the age group 0-50----. Age can be <1 year, but how can be zero?

Line 181-182: Data on patient category, smear result, TB type and HIV status were not

available for 82, 381, 27 and 67 patients, respectively. What are the numbers 82, 381, 27 and 67?

Table 1: Line 188: what is the reason for categorizing transfer in together with new and relapse case? New and relapse can be linked to DRTB and transmission trend and transfer in link to what?

Reviewer #2: 1. Paragraph one lacks references

2. Reference system is mixed one another citations in side the body of the manuscript are not uniform (on the following line numbers: L44,45,46, 49,51,54, 56,58,60,62,65,83 the authors used text citations inside in the other part they used numbering system).

3. Line 87-89... the author used old version reference and it is advisable to put the recent updated reference instead.

4. Instead of a general reference to a 5-year retrospective study, it would be clearer to specify the exact time frame, such as ‘Institutional based retrospective study on a data collected from September 2012 to August 2018.' This removes any ambiguity and gives readers a precise understanding of the data analyzed as the study used specific health center data only.

5. Line number 136-138: in line 137 its advisable to check the grammar and put ‘used’ and ‘and’ next to ‘was’ and ‘outcomes’.

6. The author used old references and its recommended to use recent edition.

6. PLOS authors have the option to publish the peer review history of their article (what does this mean?). If published, this will include your full peer review and any attached files.

Reviewer #1: No

Reviewer #2: **Yes: **Musse Girma abdela

---

## [Author Response · Author response to Decision Letter 0]

20 Mar 2024

A file containing my Responses to Academic Editor and Reviewers comments is uploaded.

---

## [Decision Letter · Decision Letter 1]

1 May 2024

Tuberculosis Treatment Outcomes and associated factors at Alemgena Health Center, Sebeta, Oromia, Ethiopia

PONE-D-23-37113R1

Dear Dr. Birri,

We’re pleased to inform you that your manuscript has been judged scientifically suitable for publication and will be formally accepted for publication once it meets all outstanding technical requirements.

Kind regards,

Frederick Quinn

Academic Editor

PLOS ONE

Additional Editor Comments (optional):

Reviewers' comments:

Reviewer's Responses to Questions

**Comments to the Author**

1. If the authors have adequately addressed your comments raised in a previous round of review and you feel that this manuscript is now acceptable for publication, you may indicate that here to bypass the “Comments to the Author” section, enter your conflict of interest statement in the “Confidential to Editor” section, and submit your "Accept" recommendation.

Reviewer #1: All comments have been addressed

2. Is the manuscript technically sound, and do the data support the conclusions?

Reviewer #1: Yes

3. Has the statistical analysis been performed appropriately and rigorously? 

Reviewer #1: Yes

4. Have the authors made all data underlying the findings in their manuscript fully available?

Reviewer #1: Yes

5. Is the manuscript presented in an intelligible fashion and written in standard English?

Reviewer #1: Yes

6. Review Comments to the Author

Reviewer #1: Now, authors substantially address all comments and I am satisfied with revision and recommended for publication

7. PLOS authors have the option to publish the peer review history of their article (what does this mean?). If published, this will include your full peer review and any attached files.

Reviewer #1: No

---

## [Editor Report · Acceptance letter]

8 May 2024

PONE-D-23-37113R1 

PLOS ONE

Dear Dr. Birri, 

I'm pleased to inform you that your manuscript has been deemed suitable for publication in PLOS ONE. Congratulations! Your manuscript is now being handed over to our production team.

Kind regards, 

on behalf of

Dr. Frederick Quinn 

Academic Editor

PLOS ONE